# Combination of Bisphenol A and Its Emergent Substitute Molecules Is Related to Heart Disease and Exerts a Differential Effect on Vascular Endothelium

**DOI:** 10.3390/ijms241512188

**Published:** 2023-07-29

**Authors:** Rafael Moreno-Gómez-Toledano, María Delgado-Marín, Sandra Sánchez-Esteban, Alberto Cook-Calvete, Sara Ortiz, Ricardo J. Bosch, Marta Saura

**Affiliations:** 1Universidad de Alcalá, Department of Biological Systems/Physiology, 28871 Alcalá de Henares, Spain; maria.delgadom@uah.es (M.D.-M.); sandrasanchez07@hotmail.com (S.S.-E.); mcooky_8@hotmail.com (A.C.-C.); sara.ortiz@edu.uah.es (S.O.); ricardoj.bosch@uah.es (R.J.B.); marta.saura@uah.es (M.S.); 2Instituto Ramón y Cajal de Investigación Sanitaria—IRYCIS, 28034 Madrid, Spain

**Keywords:** bisphenol, cardiovascular disease, endocrine disruptors, apoptosis, human cohort

## Abstract

Plastic production, disposal, and recycling systems represent one of the higher challenges for the planet’s health. Its direct consequence is the release of endocrine disruptors, such as bisphenol A (BPA), and its emerging substitute molecules, bisphenol F and S (BPF and BPS), into the environment. Consequently, bisphenols are usually present in human biological fluids. Since BPA, BPS, and BPF have structural analogies and similar hormonal activity, their combined study is urgently needed. The present manuscript studied the effect of the mixture of bisphenols (BP_mix_) in one of the world’s largest human cohorts (NHANES cohort). Descriptive and comparative statistics, binomial and multinomial logistic regression, weighted quantile sum regression, quantile g-computation, and Bayesian kernel machine regression analysis determined a positive association between BP_mix_ and heart disease, including confounders age, gender, BMI, ethnicity, Poverty/Income Ratio, and serum cotinine. Endothelial dysfunction is a hallmark of cardiovascular disease; thus, the average ratio of bisphenols found in humans was used to conduct murine aortic endothelial cell studies. The first results showed that BP_mix_ had a higher effect on cell viability than BPA, enhancing its deleterious biological action. However, the flow cytometry, Western blot, and immunofluorescence assays demonstrated that BP_mix_ induces a differential effect on cell death. While BPA exposure induces necroptosis, its combination with the proportion determined in the NHANES cohort induces apoptosis. In conclusion, the evidence suggests the need to reassess research methodologies to study endocrine disruptors more realistically.

## 1. Introduction

Plastic pollution is one of the leading environmental problems facing our society, which is currently living at a new turning point in the history of humanity, called the “Anthropocene” or the “Plastic Age” [1]. Plastic polymers have become the basis of numerous industries thanks to their versatility and affordability; therefore, their demand and productivity have maintained constant growth in recent decades. In 2019, a production volume of 368 million tons [2] was recorded, which is expected to double in just 20 years [3]. Consequently, the storage of urban waste and the continuous production and recycling of plastic polymers pose a potentially severe impact at numerous trophic levels [4], ultimately leading to chronic human exposure. Recent studies have confirmed plastic monomers’ presence in water, soil, and air worldwide [5,6,7].

One of the primary monomers studied by the scientific community, due to its environmental ubiquity and high level of detection in human biological fluids, is bisphenol A (BPA). Due to its high production volume and multiplicity of uses, BPA is considered one of the most widely used monomers in the world [8]. Thanks to its chemical properties [9], it can move throughout the human body and cross biological barriers, so it is commonly found in urine, blood, breast milk, and even amniotic fluid [7,10,11]. Furthermore, it is a known endocrine disruptor with the potential ability to affect numerous organs and systems, including the cardiovascular system [12,13,14]. The growing amount of evidence provided by the scientific community on the multiple and heterogeneous potential effects of BPA on health has made it essential to introduce new rules and regulations restricting the use of BPA.

Faced with the imperative need to replace BPA, industries have adopted structurally similar monomers, which still need to be studied in detail. The two most used molecules in the replacement process are bisphenol S (BPS) and bisphenol F (BPF) [15,16], whose presence has already been detected in water, air, and food in several parts of the world [5,17,18]. Both monomers have—like BPA—two phenols, and they differ only in their interphenolic linkers (C(CH3)2 for BPA; CH2 for BPF; SO2 for BPS) [19]. Due to their respective high degrees of structural homology, similarities have been observed in their potential endocrine disruptor abilities, which have even been described as hormonally active as BPA itself [15]. Furthermore, evidence in the academic literature suggests that BPS and BPF could exert a similar effect to BPA on adipocytes [20] and the GC-2 spermatocyte cell line [21]. In GC-2 cells, Sidorkiewicz et al. [21] manifested that the mixture of bisphenols causes effects comparable to the individual ones.

At the cardiovascular level, few works have explored the possible implications of the new BPA substitute molecules on human health. A limited amount of evidence has been developed in some cellular lines [22,23], animal models [24,25,26,27,28], and human cohorts [29,30,31]; in all of them, the perspective of the study contemplates only the individual effect of each one of the phenolic monomers. Nevertheless, it is evident that in the real world, outside the regulated laboratory setting, co-exposure to bisphenols does occur, as demonstrated in human cohorts that quantify the presence of a mixture of bisphenols in urine [32].

Consequently, the present manuscript pretends, for the first time, to study the possible cardiovascular implications of the bisphenols used mainly by the industry, considering their possible additive or synergistic effect through a retrospective cohort study of the mixture of urinary bisphenols in the NHANES cohort. After performing statistical analyses to identify links between joint exposure to these monomers and cardiovascular diseases, the effect of the mixture of bisphenols on endothelial cells will be studied using the realistic proportion of bisphenols found in the urine of subjects in the NHANES cohort.

The present work proposes two central hypotheses: first, actual exposure to the bisphenol mixture (the concentration present in the urine of the general population) is statistically related to heart disease; second, the bisphenol mixture may exert a deleterious effect on the cardiovascular endothelium, which would help to explain the cardiovascular pathological relationship with this class of endocrine disruptors. Our study will help redefine the way of studying BPA and its emerging substitute molecules and propose an update of the work methodology in investigating endocrine disruptors.

## 2. Results

### 2.1. The Retrospective Cohort Study

#### 2.1.1. Study Population

The first step in the statistical analysis of the subpopulations of the NHANES cohort was the performance of descriptive statistics. As seen in Table 1, most of the study covariates used in correcting the statistical models presented significant differences between individuals with and without the pathology of interest. Thus, individuals with heart disease presented significant differences in age, body mass index (BMI), ethnicity, poverty/income ratio, and significantly higher levels of bisphenol mixture corrected by creatinine. 

The subsequent comparative analysis of the BP_mix_ quartiles (Table 2) showed an interesting and significant increase in the percentage of HD patients. Furthermore, a significant increase in the age of the study subjects in quartiles 2, 3, and 4, compared to group 1, and a higher proportion of women was observed. In addition, a significantly lower poverty/income ratio and urinary creatinine were observed in Q2, 3, and 4. Finally, higher serum cotinine levels were determined in Q3 and Q4. The Pearson correlation coefficients showed significant associations between BP_mix_ with serum cotinine, and ΣBP_mix(USS)_ with Serum cotinine and BMI (Table 3).

#### 2.1.2. Logistic Regression Analysis

Consistent with the previously described results, the binomial logistic regression model determined a positive odds ratio with urinary BP_mix_ and ΣBP_mix(USS)_. Statistical significance remained below 0.05 even after correcting for age, gender, BMI, race/ethnicity, poverty/income ratio, and serum cotinine (and creatinine in ΣBP_mix(USS)_). In this case, due to the logarithmic normalization of the quantitative variable (BP_mix_), the results indicate that the increase of one logarithmic unit of BP_mix_ has a 1.20 (or 1.18) times greater risk of suffering heart disease, independently of covariates such as ethnicity, economic status, or smoking (Table 4). 

Next, the multinomial logistic regression model, in which the study population was subdivided into four BP_mix_ quartiles, showed statistically significant results in quartile 4 (Q4) but only in covariates groups 1 and 2. As seen in Table 5, individuals with higher urinary levels of urinary BP_mix_ have a higher risk of heart disease, independent of age, gender, and BMI, with an odds ratio of 1.54. Furthermore, gender and ethnicity significant results were also determined in the logistic regression analysis. The multinomial regression analysis determined a higher OR for women, which should be studied in future studies. Although previous work by our team has determined a possible connection with glomerular filtration [31], more scientific evidence remains. Although retrospective cohort studies do not allow the identification of cause–effect relationships between variables and pathologies, it is essential to note that they allow the identification of the risk of developing a pathology independently of covariates.

#### 2.1.3. Weighted Quantile Sum (WQS) Analysis and Bayesian kernel Machine Regression (BKMR)

In WQS regression for the mixture of bisphenols and heart disease risk, BPF exerts a greater weight on disease risk, as can be seen in Figure 1A, and the corrected analysis with the confounders confirms the result (Figure 1B). As shown in Figure 1C, the quantile g-computation analysis showed that BPS has a negative relationship with heart disease risk, while BPA and BPF have a positive effect. However, the combined effect showed a clear positive slope, demonstrating that the mixture of bisphenols in the context of a general population has a favorable risk of heart disease.

In the BKMR analysis model, the same trend was observed as in the previous analysis, although it was not statistically significant (Figure 2A). The risk of heart disease increases with increasing exposure to the bisphenol mixture, consistent with the results described in previous statistical models. When other bisphenols were fixed at the 25th, 50th, and 75th percentiles, single chemical-exposure effect analysis showed similar results as quantile g-computation analysis (Figure 2B).

The retrospective cohort study showed an interesting connection between the bisphenol mixture and heart disease risk. Since endothelial dysfunction is the first step in multiple cardiovascular pathologies, we used a proportion of bisphenols consistent with the average human exposure to explore the combined effect in endothelial cells.

### 2.2. In Vitro Study

#### 2.2.1. The Bisphenol Mixture Affects Cell Viability but Not Cell Adhesion

As shown in Figure 3A, BPA can reduce the viability of MAEC cells from 100 µM, an effect only observed in the substitute molecules at higher concentrations. When comparing the effects on the ability to alter cell viability at 200 µM, a significant difference between BPA and its derivative molecules is observed. At the highest dose (500 µM), differences between BPF and BPS were also observed, with BPS being the compound that least affected cell viability. Individual effect analysis reveals that the effect on viability is BPA > BPF > BPS.

On the other hand, the viability analysis of the mixture of bisphenols based on human exposure was realized compared with the same BPA concentration. For example, 100 µM of BPA was compared to 100 µM of BPA + 36 µM of BPS + 38 µM of BPF. In this way, the results could reveal fundamental interactions between the cell viability effects of bisphenols. Figure 3B shows that the combined effect of BPS and BPF could potentiate the deleterious effect of BPA exposure. The analysis showed significant differences at 100 µM and 200 µM. The potentiating effect of the mixture of bisphenols is coherent since structural homologies and endocrine disruptor potential activity exist in the three monomers present in biological fluids of human populations [15]. 

The subsequent cell adhesion analysis was realized to explain the difference in cell viability related to alteration in adhesion capacity in the same way we observed in previous works in cultured human podocytes [33]. However, the mixture of bisphenols does not affect the cell adhesion capacity at the concentration of 100 µM (100 + 38 + 36 µM of BP_mix_) or lower doses. As seen in Figure 3C,D, no significant changes were observed in the percentage of adherent cells in any of the experiments performed (30 and 60 min). 

#### 2.2.2. The Bisphenol Mixture Exerts a Differential Effect on Cell Death

Once significant differences between BPA and bisphenol mixture are identified, the next step in the in vitro study is the analysis of cell death by flow cytometry. The annexin V (AV) and propidium iodide (PI) assay are commonly used to identify viable, apoptotic, necrotic [34], and necroptotic cells [13]. Necroptosis is a cell death mechanism combining necrosis and necroptosis [35]. Therefore, in flow cytometry studies with AV and PI, double-positive cells are categorized as late apoptosis or necroptosis. The AV-PI flow cytometry assay results showed a significant increase in cell apoptosis (AV+) compared to the control group (Figure 4A). Interestingly, the quadrants corresponding to necrosis (AV−, IP+) or necroptosis/late apoptosis (AV+, IP+) did not differ significantly from the control group.

Cell death regulation mechanisms can be classified into two groups: caspase-dependent, such as apoptosis, and caspase-independent, such as necroptosis [36]. The relative expression of the initiator caspase caspase-8 (CASP8) and the effector caspase caspase-3 (CASP3) was analyzed to deepen the study of cell apoptosis [36]. As can be seen in Figure 4B–D, the combined exposure of phenols induced a significant increase in the expression of the active fragment of caspase-3 (17 kDa) and caspase-8 (47 kDa) at 100 µM (100 µM BPA, 38 µM BPF, and 36 µM BPS). Furthermore, the immunofluorescence assay confirmed the result observed in the Western blot of CASP3.

Interestingly, previous laboratory work carried out exclusively with BPA demonstrated that endothelial cells exposed to this monomer undergo a significant increase in the mechanisms of cell death associated with necroptosis, observing a significant increase in the expression of RIP3 associated with calcium calmodulin kinase II (CAMK II) [13].

For this reason, based on the results observed in the flow cytometry assay, we studied the relative expression of two main proteins involved in the necroptosis cell signaling pathway, RIP3 and MLKL. Consistent with the absence of changes in the number of necroptotic cells, the relative expression levels of RIP3 and MLKL were not altered after the combined treatment of bisphenols (Figure 4E,F), confirming the existence of a differential effect on the cell death between exposure to BPA and realistic exposure to BP_mix_.

## 3. Discussion

For the first time, the present manuscript has demonstrated significant associations between the bisphenol mixture and heart disease. Our initial hypothesis, based on [15,20,21], suggests that BPS and BPF could exert an additive (or synergistic) effect on BPA in the context of heart disease. Since the three phenolic molecules have a high degree of structural homology and hormonal activity (BPF and BPS have been described as hormonally active as BPA itself [15]), it is coherent to consider BP_mix_ as a single quantitative variable. Thus, the statistical analysis performed on the NHANES cohort (a **retrospective cohort study**) has yielded sufficient evidence to justify the subsequent in vitro experimental model.

Descriptive statistics and comparative analysis of the 3014 subjects in the cohort have shown the first indications of the relationship between BP_mix_ and heart disease. In addition, it has confirmed significant differences in the covariates included in the study of heart disease (gender, age, BMI, ethnicity, and poverty/income ratio. The analysis of the BP_mix_ quartiles also showed interesting statistical relationships with all the study covariates except BMI. The serum cotinine levels suggested that tobacco use is a significant source of environmental exposure to bisphenols. This effect is especially significant since bisphenol enters the bloodstream directly, avoiding the classic hepatic detoxification mechanisms (glucuronidation and sulfation) traditionally related to the oral route [9].

The subsequent binomial and multinomial logistic regression models confirmed the trends described in the previous analyses, determining that bisphenol mixture is an environmental factor related to the risk of developing heart disease, independent of other factors such as age, gender, BMI, ethnicity, poverty/income ratio, and serum cotinine. Statistical evidence does not shed light on the possible causal relationship between the variables; it only allows for determining a relative risk or odds ratio [37]. The greatest weakness of retrospective cross-sectional studies is their inefficiency in determining causal relationships. However, the high and heterogeneous number of individuals allows for the development of a robust base for future translational studies. Of the total number of individuals in the 2013–2014 and 2015–2016 NHANES cohorts, 3014 subjects were included in the study because they had all the necessary data for the correct development of the statistical study. Furthermore, WQS, quantile g-computation, and BKMR analysis showed interesting results that reaffirm the positive association between the mixture of bisphenols and heart disease risk. 

Consequently, the retrospective cohort study allowed us to delve deeper into the issue by developing an in vitro experimental model on the primary culture of murine aortic endothelial cells. The most exciting element of the present manuscript is the use of realistic proportions of a bisphenol mixture mimicking human exposure. For the first time in the academic literature, an endocrine disruptor study was contextualized in the real world to extrapolate the results with new potential human risks.

The subsequent **in vitro experimental model** demonstrated that exposure to a mixture of bisphenols in a proportion analogous to the average of a human cohort induces an effect different from BPA alone. Previous work from our laboratory linked murine endothelial cell exposure to BPA with necroptosis, a mechanism of programmed necrosis cell death [13]. However, the evidence presented in this article demonstrates that the mixture of bisphenols induces apoptosis. Beginning with the MTT viability cell assay, the individual comparative study showed that BPS and BPF have a lower ability to affect cells than BPA. This phenomenon is consistent with the work of Prudencio et al. [38], who observed that BPS and BPF are less disruptive to cardiac electrophysiology. In other cellular contexts, Castellini et al. [39] work demonstrated that BPS and BPF are safe alternatives for sperm biology. Russo et al. [40] also observed this pattern and partially attributed it to phospholipid affinity, implying that toxicity would increase with increasing membrane affinity. However, there are also certain contradictions in the literature, such as in the work of Ji et al. [23]. They observed that BPF could induce higher vascular toxicity and oxidative stress than BPA and BPS.

The results of MTT at higher concentrations showed significant differences between BPS and BPF, showing that BPS is the compound with the least capacity to modulate cell viability in murine endothelial cells. In human hepatocarcinoma cells and peripheral blood mononuclear cells, it has been observed that BPS is the phenol that generates less genotoxicity [41,42]. However, there are studies with BPS in animal models that have related it to atherosclerotic cardiovascular disease [43] and cardiotoxicity [44] in zebrafish, and altered cardiac function in mice [27] and rats [26].

When studying the combined effect, the results show a more significant effect than individual BPA. It is essential to highlight that in the comparative study, we used a fixed concentration of BPA and the corresponding proportionality determined in the NHANES cohort. For example, the concentration of 100 µM of BPA was compared with a mixture that includes 100 µM of BPA + 38 µM of BPF + 36 µM of BPS. In this way, it is easy to determine if the bisphenols added to the mixture act as agonists or antagonists. In this case, the results show a potentiating effect of the mixture of urinary bisphenols compared to the individual effect. This fact defined the cellular experimental model, using doses of 1 to 100 µM of BPA and its equivalent proportions of BPS and BPF. 

Cell adhesion could not explain the alterations in the viability assays since no changes induced by any BP_mix_ concentrations were observed. However, these preliminary studies focused on short-term exposure to the mixture of bisphenols (24 h) and we cannot exclude the possibility that chronic exposure to BP_mix_, the same type of exposure that occurs in the human population, may alter cell adhesion [45]. Indeed, previous work in our laboratory determined that BPA can substantially affect the cell adhesion capacity (as well as the structural integrity) of the human podocyte [33]. 

The flow cytometry assay showed that the mixture of bisphenols induces an increase in cell apoptosis that was not observed in the quadrant associated with necroptosis, showing an essential difference with exposure to BPA alone. In the cardiovascular context, previous work realized by our team determined that BPA exposure could promote cell death by necroptosis through CAMKII [13]. Nomura et al. [46] observed that increased concentrations of cytoplasmic calcium promote necroptosis through the activation of CAMKII, which is coherent with evidence that relates BPA exposure with alterations in intracellular calcium [47], ionic channels [48,49], or even intestinal absorption ability [50]. However, the evidence in the present manuscript strongly suggests a differential effect promoted by the realistic mixture of bisphenols.

It has been described that CASP8 can inactivate the TNFRSF1A complex activity by cleaving necroptotic protein RIP1 and, consequently, favor apoptosis through the activation of CAS3 or CAS7 [36]. The Western blot and immunofluorescence analysis showed an increased expression of both CAS8 and CAS3, confirming the significant increase in cell apoptosis observed by flow cytometry. Furthermore, the relative expression of crucial proteins implicated in necroptosis, RIP3, and MLKL, was analyzed by Western blot. As expected, no changes were observed in the assays, confirming that BP_mix_ could promote cell death through different cellular mechanisms better than BPA alone.

In previous studies, we found that inflammation and oxidative stress in BPA induced cardiac effects and as a consequence of endothelial necroptosis [14]. We have not tested the inflammatory potential of BP_mix_. Consequently, further work on future cellular and animal study models will allow us to demonstrate the molecular mechanisms underlying the new pathological evidence related to the BPA mixture.

The results described in the present work are novel due to the focus on combining bisphenols. In the literature, it is usual to find works developed in the context of a single molecule; however, the approach developed in the NHANES cohort, with the combined “real” exposure, coupled with the in vitro model proportional to the translational model, has provided completely novel evidence. Thus, despite the reduced effect on cell viability observed with BPS and BPF, individually, their combination with BPA in the human mean proportionality produces a potentiating effect. Furthermore, the observed change in the type of cell death associated with BPA or BP_mix_ opens a new avenue for studying endocrine disruptors.

The clinical applicability of the findings is fundamentally related to the type of materials commonly used in the clinical context. Due to the physical properties of plastic polymers, they are commonly used in the clinical environment. Our results demonstrate the need to replace polymers derived from bisphenols or to limit their use as enhancers in the plastics industry. 

The results of the present work fundamentally represent the breaking of two paradigms: firstly, it proposes the need to modify the methodological approach related to the study of endocrine disruptors individually; and secondly, it implies an impact on the current system of plastics manufacturing. Companies are beginning to replace BPA with monomers such as BPS or BPF; however, the evidence points to the need to apply the precautionary principle. Today, many products are marked as “BPA-free”. It is commonly found in most plastic products related to breastfeeding and newborns. However, the new substitute molecules likely do not provide a health benefit, as they could pose a risk comparable to that of BPA itself. The heterogeneity of uses of bisphenols is a problem for the consumer since they are found in most food packaging in varying proportions, as well as in everyday objects and even sanitary equipment. As a result, it is impossible to avoid contact with them. However, consumers must know they can replace plastic packaging and containers with glass. Consumers must not heat plastic containers due to the increased release of endocrine disruptors by heat. For example, using glass bottles in the baby’s early life stages could significantly reduce exposure to bisphenols.

## 4. Materials and Methods

### 4.1. The Retrospective Cohort Study

#### 4.1.1. Data Extraction from the NHANES Cohort

First, all the data from the NHANES cohort quantifying the three bisphenols used mainly in the industry (BPA, BPF, and BPS) were extracted: datasets 2013–2014 and 2015–2016. The files (.XPT) were extracted from the official website of the American Centre for Disease Control (CDC, 2016) (accessed on 1 September 2022, [51]), obtaining a total of 20,146 study subjects. Of these, 5333 individuals had quantified values of urinary bisphenols, of which 3701 were ≥18 years old. Once the data of the adult individuals with quantified levels of urinary bisphenols were obtained, the study population was subdivided according to heart disease (Figure 5). 

HD-positive subjects were defined as all individuals who answered affirmatively to any of the following questions related to heart disease on the NHANES questionnaires [52]: “Has a doctor or other health professional ever told you that you had” (a) “congestive heart failure?” (b) “(…) coronary heart disease?” (c) “(…) angina, also called angina pectoris?” (d) (…) “a heart attack (also called myocardial infarction)?” (e) “(...) a stroke?”. Of the 3701 adults studied, 3014 had all the necessary parameters for inclusion in the study group.

#### 4.1.2. Statistical Analysis

A retrospective cohort study was performed to identify possible links and patterns between heart disease and the mixture of urinary bisphenols, or BPmix. First, “Unsupervised summary scores” were performed, following the work of Li et al. [53], calculating the molar sum of bisphenols (ΣBP_mix_) by dividing each metabolite concentration by its molecular weight and then summing:$$\Sigma {BP}_{mix(USS)}=(\frac{\left[BPA,\frac{ng}{mL}\right]}{228.29\;g\times mol}+\frac{\left[BPS,\frac{ng}{mL}\right]}{250.27\;g\times mol}+\frac{\left[BPF,\frac{ng}{mL}\right]}{200.23\;g\times mol})$$

Secondly, the sum of the concentration of bisphenols corrected by the value of urinary creatinine (Ur. Creat.) was used, to avoid errors in the results due to differences in the glomerular filtration capacity of each study subject:$${B\mathrm{P\_}C}_{mix}=(\frac{\left[BPA,\frac{ng}{mL}\right]}{Ur.Creat.,\frac{mg}{dL}}+\frac{\left[BPS,\frac{ng}{mL}\right]}{Ur.Creat.,\frac{mg}{dL}}+\frac{\left[BPF,\frac{ng}{mL}\right]}{Ur.Creat.,\frac{mg}{dL}})$$

The method of bisphenol quantification uses online solid phase extraction coupled with high-performance liquid chromatography and tandem mass spectrometry (online SPE-HPLC-Isotope dilution-MS/MS). The detection limit was 0.2, 0.2, and 0.1 for BPA, BPF, and BPS, respectively. Details on protocols are available on the NHANES website [54,55].

Basic descriptive statistics were performed, analyzing the variables of interest in each of the pathological subgroups, as well as in the BPmix quartiles. Quantitative variables were described as the arithmetic mean (standard deviation), AM (SD), or as the geometric mean (95% confidence interval), GM (95% CI), depending on their normality. GraphPad Prism 7.0 software (GraphPad Software Inc., San Diego, CA, USA) was used for descriptive statistical analysis, comparative analysis, and graphical representation. The data distribution was analyzed using the D’Agostino–Pearson, Shapiro–Wilk, and Kolmogorov–Smirnov normality tests. Subsequently, T-student or Mann–Whitney test was used for the comparative analysis of two variables. The one-way ANOVA or Kruskal–Wallis followed by a Bonferroni or Dunn’s test was carried out for three or more variables. In the case of the analysis of dichotomous variables in descriptive statistics, Fisher’s exact test was used. The *p*-values in the figures and tables correspond to the post hoc test. *p* < 0.05 was considered statistically significant.

Next, a graphical representation of the BP_mix_ in HD was realized, and the Cochran q test was performed to analyze the pathology in each BP_mix_ quartile. A binomial and multinomial logistic regression study model was realized in the next step. The logistic regression analysis’s objective was to determine patterns that would identify possible relationships between the pathological subgroups and the mixture of urinary bisphenols due to the multifactorial origin of the study pathologies. Finally, each of the analyses was carried out in 3 different ways, selecting anthropometric and demographic variables, according to academic literature [53,56,57] individually (1); corrected for age, gender, and BMI (2); and corrected for (2) + race/ethnicity, poverty/income ratio, and serum cotinine (3). Urinary concentrations of bisphenol metabolites were all log-transformed to normalize distributions. The IBM SPSS Statistics for Windows software, version 27 (IBM Corp, Armonk, NY, USA), was used for the Cochran q test and regression models.

Subsequently, the weighted quantile sum (WQS) was developed with the R package “gWQS” [58] to analyze relationships between the exposures and the outcome while summarizing the complex exposure to the mixture of interest [59,60]. Briefly, the WQS regression summarizes the overall exposure to the mixture by estimating a single weighted index and accounts for the individual contribution of each component using weights [61]. 

Next, a quantile G-computation model was performed with R package “qgcomp” [62] to estimate the overall effect of bisphenols mixture with Heart Disease. The QG-comp method estimates the effect of the bisphenol mixture on the outcome by quantifying the change in outcome for each quantile increase in the concentration of all bisphenols in the mixture. It also calculates the relative contribution of each bisphenol to the overall effect, whether it has a positive or negative direction. The estimated overall mixture effect, denoted by ψ, represents the change in the outcome associated with a quantile increase in the concentration of all bisphenols in the mixture. In other words, it captures the joint effect of all the bisphenols in the mixture on the outcome. Therefore, ψ can be used to assess the overall impact of the bisphenol mixture on the outcome of interest. 

Additionally, Bayesian kernel machine regression (BKMR) was further applied, using the “bkmr” package in R [63], to account for simultaneous exposures to multiple concurrent exposures. BKMR is a novel approach that can be used to study the effects of pollutant mixtures on health outcomes. This method involves regressing the health outcome on a flexible function of the mixture components, specified using a kernel function. In high-dimensional settings, where the number of mixture components is large, BKMR incorporates a hierarchical variable selection approach. This approach helps to identify essential mixture components while accounting for the correlated structure of the mixture. By including this hierarchical variable selection approach, BKMR can handle situations with many potential mixture components, and the relationships between the components and the outcome are complex and interdependent [64,65]. In our case, let X be the vector of the mixture of components (BPA/Creat., BPS/Creat., and BPF/Creat.), Y the vector of “Heart Disease”, and Z the vector of potential confounders (same covariates selected in the logistic regression analyses); 300 knots and 3000 iterations were selected in the statistical analysis. BKMR methodology was developed according to other publications [53,66]. 

#### 4.1.3. Calculation of the Average Proportion of Bisphenols in the Human Population (Used in the Subsequent In Vitro Study)

All available data from all subjects presenting with all three urinary bisphenols were selected, obtaining a total of 5333 subjects. Next, a descriptive statistical analysis was performed. Since the BPs do not present a normal distribution, the geometric mean value was used, and the proportion of each one was calculated using the compound with the highest value (BPA) as a reference (Table 6).

### 4.2. In Vitro Study

#### 4.2.1. Cell Culture

As previously reported, Murine Aortic Endothelial Cells (MAECs) were isolated from mouse aorta [13,67]. MAECs were selected by their ability to express the intercellular adhesion molecule-2 (ICAM-2) protein and purified with a flow cytometry cell sorter (DAKO). Purification was verified by confocal microscopy of MAECs double stained with von Willebrand factor antibodies. 

MAEC were cultured with Dulbecco’s Modified Eagle’s Medium (DMEM/F12), supplemented with 0.05 mg/mL penicillin/streptomycin, 2.5 μg/mL amphotericin, and 10% Fetal Bovine Serum (Merck) in a humidified CO2 incubator with 5% CO2 at 37 °C. MAECs were used between passage 4 and 9. 

BPA, BPS, and BPF concentrations between 100 and 500 µM were used to delimit the cytotoxicity on the MTT assay (and their respective proportions in the bisphenol mixture). Subsequently, the concentration range of 1 to 100 µM (1 + 0.38 + 0.36, 10 + 3.8 + 3.6, and 100 + 38 + 36 µM of BPA, BPF, and BPS, respectively) was used to explore cell adhesion and death.

#### 4.2.2. MTT Cell Viability Assay

MAECs were grown on 24-well plates (1500 cells/well) in a complete medium. After overnight incubation, the medium was removed, and 1 mL of growth culture containing different concentrations of BPs ranging from 0 (Control) to 500 μM was added for 24 h (and their respective proportions in the phenol mixture). After BPs treatment, 100 µL of MTT (5 mg/mL) was added to each well in 900 µL of the medium, and the plates were incubated for 3.5 h at 37 °C. Afterward, the supernatant was removed, and DMSO (Sigma Aldrich, Burlington, Massachusetts, United States) was added to solubilize the formazan crystals. The absorbance was measured at a test wavelength of 570 nm [68].

#### 4.2.3. Cell Adhesion Assay

After BPs mixture treatment for 24 h in p100 plates, the medium was removed, and cells were trypsinized. Hereafter, 40,000 cells were left to settle in 24-well plates with DMEM/F12 medium containing 10% FBS for 30 and 60 min. After removing the medium, the 24-well plate was washed with PBS twice, and attached cells were measured by staining with violet crystal and subsequent cell count. Four images per condition were counted, each being duplicated [69].

#### 4.2.4. Flow Cytometry

An annexin V (AV)–propidium iodide (PI) assay was performed to identify the type of cell death associated with the treatment with the mixture of BPs; 6-well plates with 60,000 cells treated for 24 h were used. After incubation, the culture media were collected, and the cells were washed with PBS twice, trypsinized, and neutralized with the culture media collected previously. The cells were centrifuged at 1500 rpm for 5 min, and the resulting pellet was resuspended in 150 μL of 1× binding buffer, incubated with annexin V (1:50) for 30 min, and with IP (5 μL) for 15 min (Biosciences, NJ, USA). Data acquisition was carried out with the MACSQuant 10 flow cytometer (Miltenyi Biotec, Bergisch Gladbach, Germany), and its analysis was performed with the MACSQuantify program. The data were represented as a relative percentage using the control group as a reference.

#### 4.2.5. Western Blot

Protein lysates were immunoblotted as previously described [70]. Total protein was separated in SDS-polyacrylamide gel electrophoresis and transferred to a PDVF membrane. For protein detection, blocked membranes were incubated with specific antibodies, washed, and incubated with a secondary antibody. Immunoreactive bands were visualized with the ECL system (GE Healthcare Life Sciences, Chicago, Illinois, USA). 

Primary antibodies used: RIP3 (Santa Cruz, ref. sc-374639, 1:1000), MLKL (Santa Cruz, ref. sc-293201, 1:500), Cleaved Caspase-3 (Cell Signaling Technology, ref. 9664, 1:500), Caspase 8 (Cell Signaling Technology, ref. 9746, 1: 500), and GAPDH (Merck, ref. MAB374, 1:1000).

#### 4.2.6. Confocal Microscopy

Slides containing cells were incubated with the primary antibodies overnight at 4 °C (Caspase3p11 from Santa Cruz (ref. sc-271759, 1:50)). After washing with PBS, the slides were incubated with Alexa-488-conjugated secondary antibody for 1 h at room temperature. Nuclei were stained with Hoechst (1:2000). Images were taken for data quantification using a Leica TCS SP5 confocal microscope (UAH-NANBIOSIS-CIBER-BNN). At least five different fields per condition were obtained.

#### 4.2.7. Statistical Analysis

All results were expressed as mean ± Standard Deviation (SD). GraphPad Prism 7.0 software (GraphPad Software Inc., San Diego, CA, USA) was used for statistical analysis. First, the data distribution was analyzed using the D’Agostino–Pearson, Shapiro–Wilk, and Kolmogorov–Smirnov normality tests. Subsequently, one-way ANOVA or Kruskal–Wallis, followed by a Bonferroni or Dunn’s test, respectively, was carried out. In MTT and flow cytometry assays, two-way ANOVA was performed, followed by Tukey’s multiple comparisons test, Sidak’s multiple comparisons test, or Dunnett’s multiple comparisons test, as appropriate. The *p*-values in the figures and tables correspond to the post hoc test. *p* < 0.05 was considered statistically significant. All assays were performed at least three times in duplicate.

## 5. Conclusions

In conclusion, the retrospective cohort study combined with the in vitro study suggests the need to study endocrine disruptors in a more realistic context, overcoming the obsolete “one compound–one pathology” paradigm. The present manuscript has shown that the bisphenol combination can exert a different effect than BPA alone. It is imperative to reformulate the research methodology on the possible effects of endocrine disruptors since their potential interaction could radically change the effect on health. Future studies should delve into the new possibilities suggested by the results of this manuscript, combining bisphenols and other endocrine disruptors commonly present in human biological fluids. Consequently, using new BPA substitute molecules poses new challenges for research and potential dangers for all trophic levels, humans, and future generations.

## Figures and Tables

**Figure 1 ijms-24-12188-f001:**
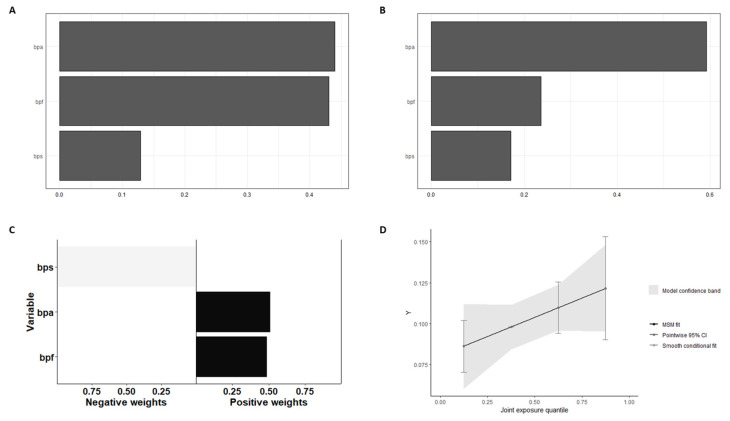
Weighted quantile sum and quantile G-computation analysis of urinary BPmix in Heart Disease: (**A**) Weights from weighted quantile sum regression for the mixture of bisphenols and HD risk, without confounders adjust. (**B**) Weights from weighted quantile sum regression for the mixture of bisphenols and HD risk, adjust the positive WQS regression model for confounders. (**C**) Weights corresponding to the proportion of the positive or negative partial effect per chemical in the quantile g-computation model. (**D**) Model fitting using Bootstrap. Note that the number of iterations was 200. The graph of this model estimates the general effect of the mixture.

**Figure 2 ijms-24-12188-f002:**
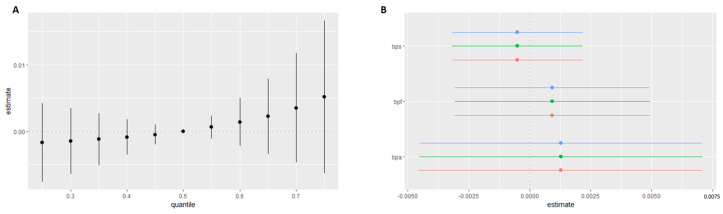
Bayesian kernel machine regression analysis of urinary BPmix in Heart Disease: (**A**) Joint effect of BPA, BPS, and BPF mixture on “Heart Disease” calculated by the BKMR model. (**B**) Single chemical-exposure effect (95% CI) to “Heart Disease” when other chemicals were fixed at a specific quantile (25th, 50th, 75th).

**Figure 3 ijms-24-12188-f003:**
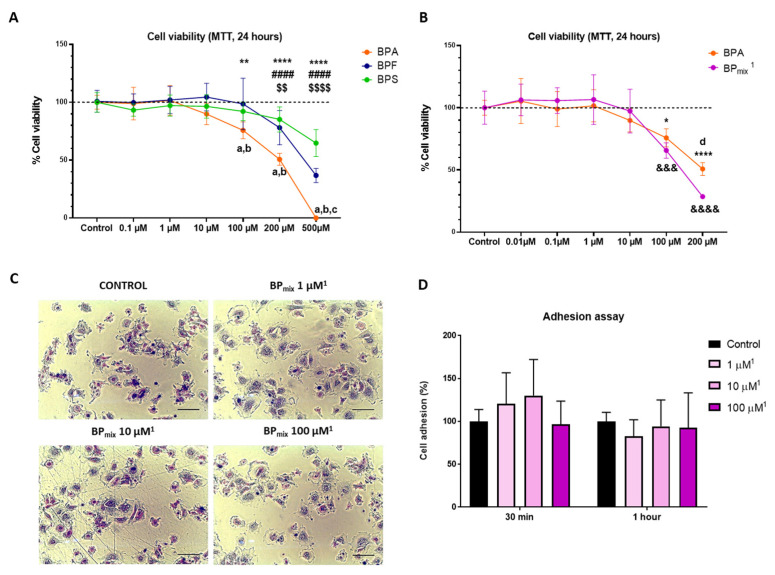
Graphic representation of exposure to PBs on cell viability (MTT assay) and cell adhesion: (**A**) Cell viability effect of individual exposure. (**B**) Comparison between the individual effect of BPA and the combined effect of the three bisphenols (BP_mix_). Two-Way ANOVA followed by Tukey’s multiple comparisons test, Sidak’s multiple comparisons test, or Dunnett’s multiple comparisons test was performed, as appropriate. (**C**) Representative photomicrographs of the adhesion test at 60 min. Scale bar = 100 µm. (**D**) Graphic representation of cell adhesion assay. The graph includes the average values of the results of the adhesion tests at 30 and 60 min. ^1^ The concentrations expressed in the figures indicate the concentration of BPA used. In addition, the corresponding proportions of BPF and BPS were added based on average data obtained from the NHANES cohort: BPA (1): BPF (0.38): BPS (0.36). The results were represented as mean (standard deviation). * represents a significant difference between the BPA group and its respective control (* *p*-value ≤ 0.05, ** *p*-value ≤ 0.01, **** *p*-value ≤ 0.0001). # represents a significant difference between the BPF group and its respective control (#### *p*-value ≤ 0.0001). $ represents a significant difference between the BPS group and its respective control ($$ *p*-value ≤ 0.01, $$$$ *p*-value ≤ 0.0001). & represents a significant difference between the BPA + BPF + BPS group and its respective control (&&& *p*-value ≤ 0.001, &&&& *p*-value ≤ 0.0001). The letter “a” indicates a significant difference between BPA and BPF; “b” between BPA and BPS; “c” between BPF and BPS; “d” between BPA and BPA + BPF + BPS.

**Figure 4 ijms-24-12188-f004:**
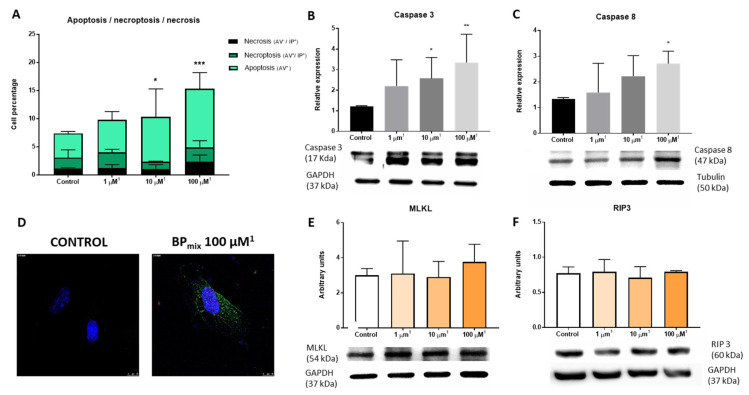
Cell death study by flow cytometry, Western blot, and immunofluorescence: (**A**) Flow cytometry: annexin V (AV) and propidium iodide (PI) assay. Two-Way ANOVA was performed, followed by Tukey’s multiple comparisons test, Sidak’s multiple comparisons test, or Dunnett’s multiple comparisons test, as appropriate. (**B**) Western blot of Caspase-3. (**C**) Western blot of Caspase-8. (**D**) Immunofluorescence of caspase-3. Scale bar = 10 µM. (**E**) Western blot of MLKL. (**F**) Western blot of RIP3. Kruskal–Wallis, followed by a Dunn’s test, was performed. The results were represented as mean (standard deviation. * *p*-value ≤ 0.05; ** *p*-value ≤ 0.01; *** is equivalent to a *p*-value ≤ 0.001. ^1^ The concentrations expressed in the graphical representation indicate the concentration of BPA used. In addition, the corresponding proportions of BPF and BPS were added based on average data obtained from the NHANES cohort: BPA (1): BPF (0.38): BPS (0.36).

**Figure 5 ijms-24-12188-f005:**
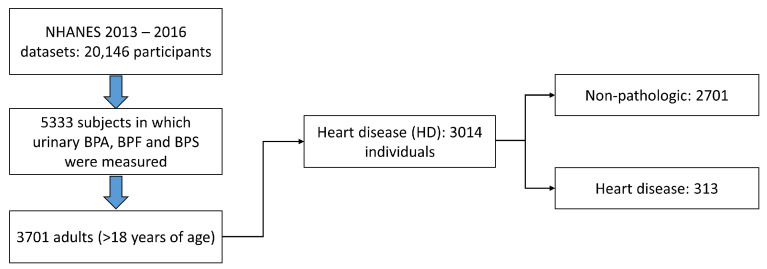
Diagram of the study subject selection process. Each one contains all the values of urinary bisphenols and the covariates data used to correct the performed statistical analyses.

**Table 1 ijms-24-12188-t001:** Descriptive analysis of the mixture of urinary bisphenols and study covariates based on heart disease. In each subgroup, the sample size is indicated in parentheses. Quantitative parameters are described as the arithmetic mean (standard deviation) or geometric mean (95% Confidence Interval) based on normality, and the dichotomous variables are represented as percentages. Statistical analysis of quantitative variables was performed using the *T*-test or Mann–Whitney test, as appropriate, and Fisher’s exact test was applied to dichotomous variables.

	Heart Disease		
Variable	Healthy (*n* = 2701)	HD (*n* = 313)	*p*-Value
Gender, men (%)	1236 (45.76)	171 (54.63)	**0.003**
Age	43.96 (43.33–44.59)	64.25 (62.71–65.84)	**0.000**
BMI (kg/m^2^)	28.47 (28.23–28.72)	30.12 (29.36–30.91)	**0.000**
Mexican American	409 (15.14)	34 (10.86)	**0.000**
Other Hispanic	299 (11.07)	27 (8.63)	
Non-Hispanic White	1000 (37.02)	144 (46.01)	
Non-Hispanic Black	582 (21.55)	80 (25.56)	
Other Race—Including Multi-Racial	411 (15.22)	28 (8.95)	
Poverty/Income Ratio	1.90 (1.83–1.96)	1.56 (1.41–1.73)	**0.000**
Serum cotinine (mg/dL)	0.25 (0.22–0.29)	0.30 (0.19–0.47)	0.36
Urinary creatinine (mg/dL)	101.2 (98.42–104.1)	96.7 (89.64–104.3)	0.076
BP_C_mix_ (ng/g creat.)	2.9 (2.80–3.00)	3.41 (3.06–3.79)	**0.004**
ln-BP_C_mix_ (ng/g)	1.06 (0.90)	1.23 (0.97)	**0.004**
ΣBP_mix_ (nM)	12.96 (12.44–13.50)	14.62 (12.85–16.62)	0.101
ln-ΣBP_mix_ (nM)	2.56 (1.09)	2.68 (1.15)	0.101

Abbreviations: BMI, Body Mass Index; BPmix, the mixture of urinary bisphenols; Creat, Creatinine. Bold: statistically significant values

**Table 2 ijms-24-12188-t002:** Descriptive analysis of covariates of interest based on the quartile of urinary bisphenols. In each subgroup, the sample size is indicated in parentheses (*n*). Quantitative parameters are described as geometric mean (95% Confidence Interval), and dichotomous variables are represented in percentages. Statistical analysis of quantitative variables was performed using the Kruskal–Wallis followed by Dunn’s test, and Fisher’s exact test was applied to dichotomous variables. * *p*-value ≤ 0.05; ** *p*-value ≤ 0.01; **** *p*-value ≤ 0.0001. # *p*-value ≤ 0.05 vs. Q2; #### *p*-value ≤ 0.0001 vs. Q2. Note that Fisher’s exact test performed on diabetes in quartile 4 has obtained a *p*-value of 0.056).

Variable	Q1 (*n* = 750)	Q2 (*n* = 752)	Q3 (*n* = 757)	Q4 (*n* = 755)	*p*-Value (Chi-Square)
Heart Disease (%)	**60 (8.0)**	**75 (9.97)**	**82 (10.83)**	**96 (12.72)**	**0.026**
Gender, men (%)	**441 (58.8)**	**333 (44.34)**	**314 (41.53)**	**319 (42.25)**	**0.000**
Age	43.63 (42.43-44.86)	**46.95 (45.71–48.22) ***	**45.83 (44.56–47.14) ****	**46.55 (45.29–47.84) ****	
BMI (kg/m^2^)	28.38 (27.93–28.84)	28.47 (28.02–28.92)	28.83 (28.36–29.3)	28.88 (28.41–29.36)	
Mexican American	**111 (14.8)**	**115 (15.31)**	**116 (15.34)**	**101 (13.38)**	**0.000**
Other Hispanic	**70 (9.33)**	**84 (11.19)**	**92 (12.17)**	**80 (10.60)**	
Non-Hispanic White	**258 (34.40)**	**281 (37.42)**	**302 (39.95)**	**303 (40.13)**	
Non-Hispanic Black	**153 (20.40)**	**168 (22.37)**	**158 (20.90)**	**183 (24.24)**	
Other Race—Including Multi-Racial	**158 (21.07)**	**104 (13.85)**	**89 (11.77)**	**88 (11.66)**	
Poverty/Income Ratio	2.08 (1.96–2.21)	**1.80 (1.69–1.93) ***	**1.79 (1.68–1.92) ****	**1.78 (1.67–1.90) ****	
Serum cotinine (mg/dL)	0.15 (0.12–0.20)	0.21 (0.16–0.28)	**0.31 (0.23–0.41) ****	**0.43 (0.32–0.58) **** ####**	
Urinary creatinine (mg/dL)	120.4 (115.1–125.9)	**102.1 (96.99–107.4) ******	**89.56 (84.73–94.66) **** #**	**93.67 (88.62–99.0) ******	

Abbreviations: BMI, Body Mass Index; HbA1c, glycosylated hemoglobin A1c; BPmix, the mixture of urinary bisphenols. Bold: statistically significant values

**Table 3 ijms-24-12188-t003:** Correlation analysis (Pearson correlation coefficient) of quantitative variables. * *p*-value ≤ 0.05; *** *p*-value ≤ 0.001; **** is equivalent to a *p*-value ≤ 0.0001.

	BP_C_mix_ (ng/g creat.)	ΣBP_mix_ (nM)	Age	BMI (kg/m^2^)	Serum Cotinine (mg/dL)	Poverty-Income Ratio
**BP_C_mix_ (ng/g creat.)**	**1**	********	-	-	*****	-
**ΣBP_mix_ (nM)**	**0.824**	1	-	*****	*****	-
**Age**	0.023	−0.019	1	-	*******	********
**BMI (kg/m^2^)**	0.012	**0.041**	0.016	1	-	********
**Serum cotinine (mg/dL)**	**0.044**	**0.042**	**−0.060**	−0.034	1	********
**Poverty/Income Ratio**	0.016	0.023	**0.06**	**−0.089**	**−0.175**	1

Bold: statistically significant values

**Table 4 ijms-24-12188-t004:** Association between the mixture of urinary bisphenols and Heart Disease risk (odds ratio). Each of the binomial logistic regression analyses was performed in 3 different ways: individually (1); corrected for age, gender, and BMI (2); and corrected for (2) + race/ethnicity, poverty/income ratio, and serum cotinine (3); in ΣBP_mix(USS),_ urinary creatinine was included in the covariate group 3. Urinary concentrations of bisphenol metabolites were all log-transformed to normalize distributions.

BP Mixture	Covariates	OR (95% CI)	*p*-Value
BP_C_mix_ (ng/g creat.)	1	**1.20 (1.06–1.35)**	**0.003**
	2	**1.20 (1.05–1.37)**	**0.007**
	3	**1.20 (1.05–1.38)**	**0.008**
ΣBP_mix_ (nM)	1	1.10 (0.99–1.22)	0.066
	2	**1.14 (1.02–1.28)**	**0.023**
	3	**1.18 (1.04–1.35)**	**0.013**

Abbreviations: BP, Bisphenol; OR, Odds Ratio; CI, Confidence Interval. Bold: statistically significant values

**Table 5 ijms-24-12188-t005:** Multinomial logistic regression between Heart Disease and the mixture of bisphenols quartile. Note that quartile 1 is the reference group for the statistical study model. Each of the binomial logistic regression analyses was performed in 3 different ways: individually (1); corrected for age, gender, and BMI (2); and corrected for (2) + race/ethnicity, poverty/income ratio, and serum cotinine (3).

Mixture BPs Quartile	Covariates	OR (95% CI)	*p*-Value
BP_C_mix_ Q1	1	REF	-
	2	REF	-
	3	REF	-
BP_C_mix_ Q2	1	1.27 (0.89–1.82)	0.182
	2	1.12 (0.77–1.63)	0.568
	3	1.06 (0.72–1.56)	0.761
BP_C_mix_ Q3	1	1.40 (0.98–1.98)	0.061
	2	1.30 (0.90–1.89)	0.162
	3	1.24 (0.85–1.81)	0.263
BP_C_mix_ Q4	1	**1.68 (1.19–2.35)**	**0.003**
	2	**1.54 (1.07–2.21)**	**0.021**
	3	1.41 (0.98–2.05)	0.067

Abbreviations: REF, reference; BPs, Bisphenols; OR, Odds Ratio; CI, Confidence Interval. Bold: statistically significant values

**Table 6 ijms-24-12188-t006:** Quantitative analysis of urinary bisphenols in the human population (*n* = 5333) and their proportionality.

Compound	GM (95% CI), ng/mL	MW (g/mol)	Molarity (nM)	Proportion
Bisphenol A	1.22 (1.18–1.25)	228.29	5.34	1
Bisphenol S	0.49 (0.47–0.51)	250.27	1.96	0.36
Bisphenol F	0.41 (0.39–0.42)	200.23	2.05	0.38

Abbreviations: GM, Geometric Mean; CI, Confidence Interval; MW, Molecular Weight.

## Data Availability

Data will be made available on request.

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
