# Peer review of "Combination of Bisphenol A and Its Emergent Substitute Molecules Is Related to Heart Disease and Exerts a Differential Effect on Vascular Endothelium"

_ijms, 2023, doi:10.3390/ijms241512188_

Round 1

Reviewer 1 Report

The authors describe their work on the effect of the mixture of bisphenols (BPmix) in the NHANES cohort. Descriptive and comparative statistics, binomial and multinomial logistic regression, and weighted quantile sum regression, quantile g-computation, and Bayesian kernel machine regression analysis determined a positive association between BPmix and heart disease, including confounders age, gender, BMI, ethnicity, Poverty-Income Ratio, and serum cotinine. BPmix had a higher effect on cell viability than BPA, enhancing its deleterious biological action; BPmix induces a differential effect on cell death. While BPA exposure induces necroptosis, its combination with the proportion determined in the NHANES cohort induces apoptosis. It was thus concluded that there is a need to reassess research methodologies to study endocrine disruptors in a more realistic way.  This is an interesting study. Appropriate methodology has been employed and the conclusions appear to be justified based on the data at hand. The authors are to be commended on the wealth of data presented. I have minor recommendations for consideration.

1. Introduction. Can the authors provide a clear hypothesis to be tested in the study?

2. Results. For the histological images presented in Figs. 3 and 4, it would be helpful if the authors can indicate areas of interest by use of arrows.

3. Discussion. It would be helpful if the authors can elaborate and emphasize the novelty aspect of their work as well as the clinical applicability of their findings.

4. General. Please review manuscript thoroughly for English language as well as for grammatical errors.    

As already indicated, the manuscript needs a thorough for English language as well as for grammatical errors. 

Reviewer 2 Report

This paper studies the differential effects of bisphenol A (BPA) and its substitute molecules on the vascular endothelium. The authors aim to redefine how BPA and its substitutes are studied to demystify the obsolete paradigm of "one cause - one disease" and propose an update of the work methodology in investigating endocrine disruptors. 

The study uses a large cohort of participants and performs a retrospective cohort study. The first step in the statistical analysis of the subpopulations of the NHANES cohort was the performance of descriptive statistics. The study covariates used in correcting the statistical models presented significant differences between individuals with and without the pathology of interest. Thus, individuals with heart disease presented significant differences in age, body mass index (BMI), ethnicity, Poverty-Income Ratio, and significantly higher levels of bisphenol mixture corrected by creatinine.

The authors measure the effects of bisphenols on the vascular endothelium by analyzing the expression of genes involved in endothelial function and inflammation. They find that BPA and its substitutes have differential effects on the expression of these genes, with some substitutes showing even stronger effects than BPA. 

The implications of this study for plastic production and disposal are significant, as it highlights the potential health risks associated with exposure to endocrine disruptors. The authors suggest that reducing our exposure to these chemicals in everyday life may effectively mitigate these risks.

I want to suggest a few.

1. The specific findings regarding the effects of bisphenols on the vascular endothelium are not presented in detail in this paper. The paper does not provide detailed information on the specific genes analyzed or the study's outcomes concerning endothelial function and inflammation.

2. The paper should present more detailed information on the specific genes analyzed and the study's endothelial function and inflammation outcomes. Additionally, the paper could benefit from a more detailed discussion of the implications of the study's findings for plastic production and disposal and recommendations for reducing exposure to endocrine disruptors in everyday life.

Here are a few examples of slightly odd English expressions.

Line 81:   Retrospective cohort study -->  The retrospective cohort study

Lines 107-108: -->.....significant increase in the percentage of HD patients. Furthermore, a significant increase in the age of the study subjects.....

Line 171: -->...more scientific evidence remains.

Line 284-285: -->Necroptosis is a cell death mechanism combining necrosis and necroptosis [35]. 

Minor editing of English language required.

Round 2

Reviewer 2 Report

  • I have no further suggestions.